# Cycle Consistent Embedding of 3D Brains with Auto-Encoding Generative Adversarial Networks

**Shibo Xing**[1]                                                            SHX26@PITT.EDU

**Harsh Sinha**[2]                                                  HARSH.SINHA@PITT.EDU

**Seong Jae Hwang**[1,2]                                                    SJH95@PITT.EDU

[1]*Department of Computer Science and* [2]*Intelligent Systems Program, University of Pittsburgh*

## Abstract

Modern generative adversarial networks (GANs) have been enabling the realistic generation of full 3D brain images by sampling from a latent space prior $\mathcal{Z}$ (i.e., random vectors) and mapping it to realistic images in $\mathcal{X}$ (e.g., 3D MRIs). To address the ubiquitous mode collapse issue, recent works have strongly imposed certain characteristics such as Gaussianness to the prior by also explicitly mapping $\mathcal{X}$ to $\mathcal{Z}$ via encoder. These efforts, however, fail to accurately map 3D brain images to the desirable prior, which the generator assumes to be sampling the random vectors from. On the other hand, Variational Auto-Encoding GAN (VAE-GAN) solves mode collapse by enforcing Gaussianness by two learned parameter, yet causes blurriness in images. In this work, we show how our *cycle consistent embedding* GAN (CCE-GAN) both accurately encodes 3D MRIs to the standard normal prior, and maintains the quality of the generated images. We achieve this without a network-based code discriminator via the Wasserstein measure. We quantitatively and qualitatively assess the embeddings and the generated 3D MRIs using healthy T1-weighted MRIs from ADNI.

**Keywords:** Auto-Encoder, Latent Space, Generative Adversarial Network, Cycle Consistency, 3D MRI

## 1. Introduction

Beyond 2D brain images, generative adversarial networks (GANs) have been promisingly generating the full-slice 3D structural MRI as well (Kwon et al., 2019) by capturing the mapping between the latent space ($\mathcal{Z}$-space) and the original image space ($\mathcal{X}$-space). Although the mapping $\mathcal{Z} \rightarrow \mathcal{X}$ is of top priority towards realistic generation, the precise construction of the latent space from the dataset ($\mathcal{X} \rightarrow \mathcal{Z}$) also benefits the overall generation quality (Larsen et al., 2016). Yet, we identified that recent works do not achieve this accurately for 3D brain generation tasks. In particular, we observed that a recent work, 3D-$\alpha$-WGAN (Kwon et al., 2019) using the Code Discriminator (CD) network to promote "realistic" embeddings of 3D brains exhibited such case. Contrarily, while VAE-GAN (Larsen et al., 2016) constructs the $\mathcal{Z}$-space similar to that of random vectors (e.g., multivariate standard normal) sampled by the generator, the generated 3D brains suffer from blurriness.

In this work, we aim to improve the latent space mapping $\mathcal{X} \rightarrow \mathcal{Z}$ via two mechanisms: (1) we use the Wasserstein distance instead of the CD similar to VAE-GAN but with the improved generation quality (i.e., Wasserstein Auto-Encoding GAN), and (2) further improve the mapping with a cycle consistency loss (Zhu et al., 2017) in the $\mathcal{Z}$-space. Our Cycle Consistent Embedding GAN (CCE-GAN) demonstrates improved embedding and generation of 3D T1W MRIs of normal aging cohort from the ADNI study.

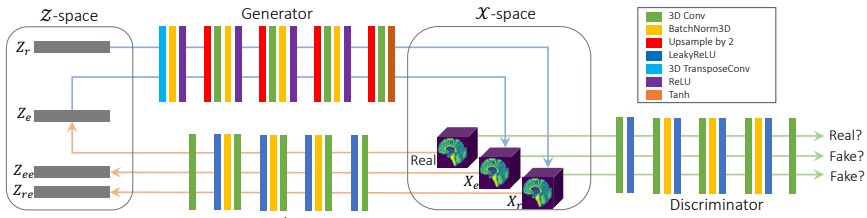

Figure 1: CCE-GAN Architecture.

## 2. Methods

The goal of GAN is to learn a generator $G$ mapping a random vector $z_r \in \mathcal{Z}$ (often a multivariate standard normal) to an image $X_r \in \mathcal{X}$ and a discriminator $D$ which differentiates a generated image $X_r$ from the real image. Due to the poor coverage of the $\mathcal{Z}$-space, randomly generated images often suffer from mode collapse. VAE-GAN (Larsen et al., 2016) addresses this issue by learning an encoder $E$ mapping from $\mathcal{X}$ to $\mathcal{Z}$ to derive the embeddings $z_e$. This results in blurry images, so a recently developed 3D-$\alpha$-WGAN (Kwon et al., 2019) leverages the CD-based encoder loss from $\alpha$-GAN (Rosca et al., 2017). Although the mode collapse issue is alleviated in the $\mathcal{X}$-space without blurriness, we identified that $E$ cannot accurately construct the $\mathcal{Z}$-space from 3D brains (e.g., $z_e \neq$ standard normal prior).

We leverage these findings to achieve improved embeddings of 3D brains, which also results in better image quality without mode collapse issue. Our solution builds on 3D-$\alpha$-GAN. First, instead of using the CD from 3D-$\alpha$-GAN or KL loss from VAE-GAN, we use the Wasserstein loss explicitly in the $\mathcal{Z}$-space between the random vectors and the embeddings. We refer to this as the Wasserstein Auto-Encoder GAN (WAE-GAN) which provides a more flexible mapping compared to the variational approach while more strictly enforcing the Gaussianness than the CD. Second, we further improve the $\mathcal{Z}$-space by deriving two additional cycle consistent embeddings: $z_{ee} = E(G(z_e))$ and $z_{re} = E(G(z_r))$. Thus, our final model, Cycle Consistent Embedding GAN (CCE-GAN), solves the following:

$$\underset{D}{\text{argmin}} \ \mathbb{E}_{z_e}[D(X_e)] + \mathbb{E}_{z_r}[D(X_r))] - 2\mathbb{E}_{x_{real}}[D(x_{real})] + \lambda_1 L_{gp}(D)$$

$$\underset{G,E}{\text{argmin}} - \mathbb{E}_{z_e}[D(X_e)] - \mathbb{E}_{z_r}[D(X_r)] + \lambda_2||X_r - X_e||_1 + \lambda_3||z_r - z_{re}||_2 + \lambda_3||z_e - z_{ee}||_2 + \mathbb{W}_l(z_r, z_e)$$

where $\mathbb{W}_l(z_r, z_e)$ is the Wasserstein loss between $z_r$ and $z_e$, and $\lambda_1, \lambda_2, \lambda_3 = 10$. $L_{gp}(D)$ is the Wasserstein gradient penalty loss for $D$ (Gulrajani et al., 2017). Note that we solely focus on improving $E$ which also improves the image generation quality as shown next.

## 3. Experiments and Conclusion

**Dataset.**    From the Alzheimer's Disease Neuroimaging Initiative (ADNI, adni.loni.usc.edu), we use 3D T1-weighted MRI of 991 CN subjects which we further processed with FreeSurfer 6.0 recon-all pipeline (surfer.nmr.mgh.harvard.edu). We further crop each scan, resize them to be of size $64 \times 64 \times 64$, and rescale the intensities to [-1,1].

**Models and Training.**    We compare the following models: (1) 3D-$\alpha$-WGAN (Kwon et al., 2019), (2) VAE-GAN (Larsen et al., 2016), (3) WAE-GAN, (4) CCE-GAN. For all models, using RTX 2080 Ti 11GB, the training involves ADAM ($\beta_1 = 0.9$, $\beta_2 = 0.999$, $lr = 0.0002$) with the mini-batch size of 4 for 100K iterations ($\approx$40 epochs). See our code for details.

**Evaluation.**    We compute the Maximum Mean Discrepancy (MMD) measure (linear and RBF kernels) between the real images and the generated images for the $\mathcal{X}$-space, and

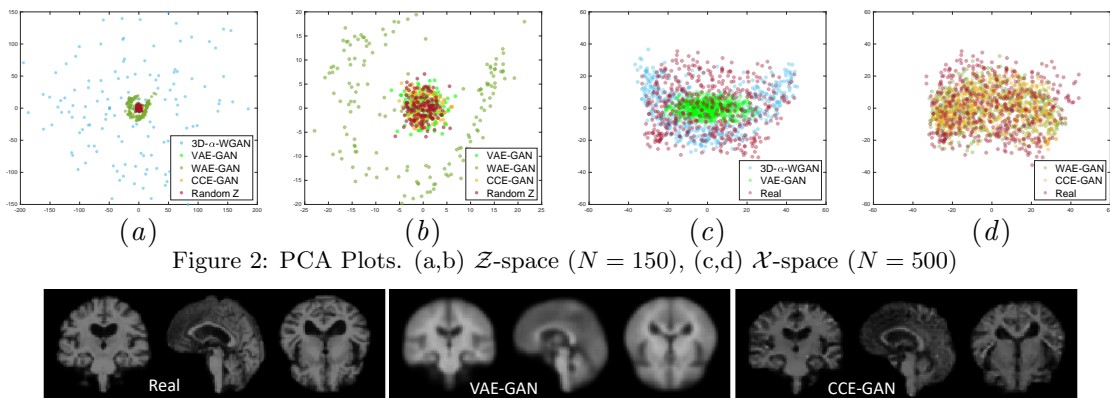

Figure 2: PCA Plots. (a,b) $\mathcal{Z}$-space ($N = 150$), (c,d) $\mathcal{X}$-space ($N = 500$)

Figure 3: Examples of Real (Left), VAE-GAN generated (Middle), and CCE-GAN generated (Right) brains.

between the random vectors $z_r$ and their corresponding embeddings $z_e$ for the $\mathcal{Z}$-space. We take the average of 100 MMDs. The Structural Similarity (SSIM) measures the distribution diversity where the real data SSIM is 0.839. For each model, we generate 1000 image pairs and compute the average SSIM which aims to be similar to the real data SSIM.

**Results: $\mathcal{Z}$-space.** We first check the encoder outputs of the real images to evaluate the embeddings compared to the random standard normal $\mathcal{Z}$-space (1000-D). Fig. 2a

| Model | $\mathcal{X}$-space | | | $\mathcal{Z}$-space | |
|---|---|---|---|---|---|
| | Linear | RBF | SSIM | Linear | RBF |
| 3D-$\alpha$-GAN | 762.4 | 0.77 | **0.841** | 619579.4 | 3.06 |
| VAE-GAN | **354.1** | 1.13 | 0.971 | 249.4 | **0.42** |
| WAE-GAN | 765.4 | 0.78 | 0.851 | 602.1 | 1.04 |
| CCE-GAN | 675.4 | **0.73** | 0.848 | **192.1** | 0.59 |

Table 1: **Linear** and **RBF** MMD, and **SSIM**.

and b show the PCA embeddings of 150 random examples. Fig. 2a shows that 3D-$\alpha$-WGAN produces sparse embeddings, while Fig. 2b shows that VAE-GAN and CCE-GAN produce embeddings highly similar to the random $\mathcal{Z}$-space also shown quantitatively in Table 1.

**Results: $\mathcal{X}$-space.** Table 1 shows the advantage of CCE-GAN, and Fig. 2d shows the PCA embeddings of the images being closer to the real dataset than Fig. 2c. In Fig. 3, we see VAE-GAN, despite reasonable MMD measures in $\mathcal{Z}$ and $\mathcal{X}$, results in blurry images.

**Conclusion.** We enable accurate mapping of 3D brains in $\mathcal{X}$-space to their embeddings in the $\mathcal{Z}$-space in a cycle consistent manner using CCE-GAN. We show that if a better embedding is achieved, it also leads to better image generation for 3D MRI generation task as well. For future work, we will consider other 3D brain datasets.

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
