# OpenReview forum: "Cycle Consistent Embedding of 3D Brains with Auto-Encoding Generative Adversarial Networks"
_MIDL.io/2021/Conference/Short — MIDL 2021 Poster_

### Official Review · Reviewer_gpVG · 2021-04-20

**Confidence:** 3
**Final Rating:** 3

**Summary:**

This paper presents an approach for 3D brain image generation. The method is based on 3D-α-GAN and newly introduces a cycle consistent regularization onto the embeddings. Experimental results show that the proposed cycle consistent regularization contributes to encode 3D MRIs to the standard normal prior and obtains better image generation results.

**Strengths:**

The motivation of the proposed method is clear. The method is simple and easy to follow. The qualitative analysis shows that the introduced cycle consistent regularization could contribute to produce embeddings more similar to the random Z-space.

**Weaknesses:**

The proposed method is based on the hypothesis that a embedding that is more similar to the desirable prior could lead to a better 3D brain image generation.  Although as shown in Fig. 2(a) and (b), the proposed CCE-GAN obtains better embedding, the image generation results in Table 1 do not show clear advantage of CCR-GAN over other methods.

**Deanonymize Review:**

no

**Justification Of The Rating:**

The proposed method is reasonable and obtains embeddings that are more similar to the random z-space for image generation, but the correlation between a better embedding and improved image generation results should be further validated.

**Paper Type:**

both

**Special Issue:**

no

---

### Official Review · Reviewer_gsZG · 2021-04-30

**Confidence:** 4
**Final Rating:** 3

**Summary:**

The authors proposed an approach to stable the training of GANs, especially the mode collapse issue. The paper show show how cycle consistent embedding GAN (CCE-GAN) can encode 3D MRIs to the standard normal prior in a cycle consistent manner in the latent space. CCE-GAN is validated on the  healthy T1-weighted MRIs from ADNI database.

**Strengths:**

1.	Figure 1 is well-done. It makes the entire method easy to understand.
2.	The paper is mostly well-written and easy to follow.
3.	The results are well presented, both in terms of quality and quantity.

**Weaknesses:**

1.	Some of the claims made by the authors are rather ambiguous. For example, in abstract the authors claimed that standard normal is the desirable prior for 3D brains images without any proper justification.
2.	The writing of the method is often difficult to comprehend. The equations could have been presented in better ways rather than just summing up with a weighting factor lambda.
3.	While this is a method driven paper, I would have liked some examples of clinical applications where it can potentially be applied.


**Deanonymize Review:**

no

**Justification Of The Rating:**

While the results section is strong, some method part needs to be clarified. Please address the major weaknesses from the previous section.

Changes in Final Version
Please address the major weaknesses from the previous section.


**Paper Type:**

methodological development

**Special Issue:**

no

---

### Meta-Review · Area_Chair_bvGj · 2021-05-09

**Recommendation:** Accept (Poster)
**Confidence:** 4

**Metareview:**

Both reviewers see some merit in this work and recommend acceptance subject to small modifications. I agree with them.

---

### Decision · Program_Chairs · 2021-05-11

Accept (Poster)